# Fixed-Dose Combination Formulations in Solid Oral Drug Therapy: Advantages, Limitations, and Design Features

**DOI:** 10.3390/pharmaceutics16020178

**Published:** 2024-01-26

**Authors:** Christi A. Wilkins, Hannlie Hamman, Josias H. Hamman, Jan H. Steenekamp

**Affiliations:** Centre of Excellence for Pharmaceutical Sciences (Pharmacen™), Faculty of Health Sciences, North-West University, Private Bag X6001, Potchefstroom 2520, South Africa; christiwilkins09@gmail.com (C.A.W.); hannlie.hamman@nwu.ac.za (H.H.); sias.hamman@nwu.ac.za (J.H.H.)

**Keywords:** fixed-dose combination therapy, formulation considerations, formulation approaches/technology

## Abstract

Whilst monotherapy is traditionally the preferred treatment starting point for chronic conditions such as hypertension and diabetes, other diseases require the use of multiple drugs (polytherapy) from the onset of treatment (e.g., human immunodeficiency virus acquired immunodeficiency syndrome, tuberculosis, and malaria). Successful treatment of these chronic conditions is sometimes hampered by patient non-adherence to polytherapy. The options available for polytherapy are either the sequential addition of individual drug products to deliver an effective multi-drug regimen or the use of a single fixed-dose combination (FDC) therapy product. This article intends to critically review the use of FDC drug therapy and provide an insight into FDC products which are already commercially available. Shortcomings of FDC formulations are discussed from multiple perspectives and research gaps are identified. Moreover, an overview of fundamental formulation considerations is provided to aid formulation scientists in the design and development of new FDC products.

## 1. Introduction

The increase in noncommunicable chronic diseases is becoming a global epidemic [1,2]. Cardiovascular diseases (CVD) and chronic respiratory diseases together with their associated complications and comorbidities have been the top two leading causes of death worldwide for the past decade [3,4]. Whilst single-drug therapy, or monotherapy, is traditionally the preferred drug treatment starting point for chronic diseases such as hypertension and diabetes, situations and disease circumstances dictate and necessitate the use of multiple drugs (polytherapy) from the onset of treatment in certain diseases such as human immunodeficiency virus (HIV), acquired immunodeficiency syndrome (AIDS), tuberculosis (TB), and malaria [5,6,7]. For infectious diseases, polytherapy is employed primarily for the prevention of resistance [8]. Alternatively, some diseases, such as CVD, require multiple drugs to handle the associated comorbidities [9]. Polytherapy may be achieved via the administration of multiple individual drug products (tablets and/or capsules) one after another, or a fixed-dose combination (FDC) product incorporating two or more drugs into a single dosage form [10].

This article intends to critically review the use of FDC therapy and provide an insight into solid oral FDC products as well as offer a comprehensive overview of the formulation considerations that need to be taken when formulating a solid oral dosage form comprising an FDC. With this work, the authors intend to provide a synopsis of the usefulness of select FDC therapies whilst recognizing their shortcomings and identifying gaps in research. Fixed-dose combination medicine products encompass various API combinations and dosage form types to be administered via different administration routes. Fixed-dose combination formulations are frequently used in the treatment of conditions and diseases including acute conditions such as influenza, pain and fever, and several infectious diseases, as well as chronic diseases such as diabetes, asthma, chronic obstructive pulmonary disease, HIV infection, and cardiovascular diseases including hypertension, congestive heart failure, and dyslipidemia. Due to the numerous FDC products available, the extent of conditions and diseases that are treated with FDC products and the availability of a number of administration routes available for FDC therapy, the topics covered in this article were intentionally delineated to cover selected conditions and diseases (representing both acute and chronic conditions and diseases). Furthermore, this article is restricted to solid FDC products (e.g., tablets and capsules) that are administered via the oral route. The purpose of the delineation is to keep the review as broad as possible to present a relevant and applicable, but not inordinately long, review of fixed-dose combinations within the scope of this article.

## 2. Fixed-Dose Combination Formulations in Drug Therapy

Fixed-dose combination formulations have been developed to combine active pharmaceutical ingredients (APIs) with different mechanisms of action into one dosage form [11]. An FDC formulation in the form of a tablet is also referred to as a polypill, which is a term that was first coined in the context of CVD prevention [12,13]. This term has subsequently become quite generic and gained broader acceptance for conditions in a wider scope. The prefix “poly”, meaning “many” or “multiple”, refers to the number of drugs included in a single dosage unit, which can be two or more. FDC therapy may be aimed at a single underlying condition or a group of commonly related conditions as this expands the pool of potential patients (critical mass) for whom a given combination of drugs would be appropriate from the prescriber’s and manufacturer’s perspectives [14].

When designing formulations, the primary consideration is the end user. We have a duty as medical professionals and medication stewards to ensure therapeutic regimens and medications are safe for patients and that the benefits of using a particular product outweigh the risks. This includes designing therapeutic regimens and products that promote safe and correct use. FDC therapies address the need for simplified, rational drug use and boast many advantages over polytherapy regimens, where individual products are taken together. However, disease-specific formulation challenges do exist. The therapeutic advantages and formulation challenges faced are first discussed in the broader context relating to FDC therapy in the next section and then from a disease-specific perspective in the section thereafter. This review of FDC therapy focuses on select solid oral FDC products employed in non-communicable diseases, namely, cardiovascular diseases and diabetes, as highly prevalent global health challenges, as well as communicable diseases such as malaria, tuberculosis, and HIV. The use of FDC therapy, however, extends much further, including use in infectious diseases (*Helicobacter pylori* infection), the treatment of endocrine and nervous system diseases (depression, Alzheimer’s disease), respiratory diseases (asthma, chronic obstructive pulmonary disease) as well as pain, influenzas, and allergy preparations.

## 3. Advantages of Fixed-Dose Combination Formulations in Drug Therapy

### 3.1. Patient Adherence

Successful treatment of chronic conditions is sometimes hampered by patient non-adherence to polytherapy. FDC therapy incorporates multiple drugs into a single dosage form to improve treatment success rates by bolstering patient adherence [15,16]. The World Health Organization (WHO) defines adherence to long-term, chronic therapy as the degree of adherence to the pharmacological treatment as well as making considerable lifestyle changes as agreed upon with the doctor and other healthcare professionals [17]. Therefore, adherence implies both the correct use of medication in accordance with the prescribed dosage and dosing frequency as well as lifestyle changes over time. Unlike compliance, it implies a patient’s active participation [17,18,19,20]. A lack of adherence to therapy is a major problem experienced globally in both developed and developing countries, and remains one of the greatest obstacles to improving individual patient health and quality of life as well as reducing the burden on health and social care systems in the private and state sectors [21,22]. Structured support in terms of treatment simplification, ongoing patient-prescriber communication, and a multidisciplinary healthcare team to provide tailor-made solutions for each patient is required.

The impact of therapeutic regimen failings extends beyond the effect on individual patient health, since it carries a huge burden described by both economic and social costs to the private and state sector health care systems as the impact of nonadherence to pharmacological therapy directly translates to higher hospitalization rates, loss of productivity, and greater incidences of death [1,15,23]. Non-compliance also translates further into an increased risk of the occurrence of drug resistance, which is of particular importance in diseases such as TB, malaria, HIV, and AIDS.

Global health campaigns need to promote the longstanding viewpoint that “prevention is better than cure”. Lifestyle choices are often the leading cause of a chronic condition developing [24]. For example, the consequence of obesity can lead to cardiac complications and/or diabetes and further progression to peripheral neuropathy. However, chronic conditions are not only lifestyle-related, and every effort needs to be made to simplify treatment regimens and bolster patient compliance to achieve therapeutic outcomes. The convenience of taking several APIs in one product as an FDC can facilitate improved compliance by the patient with the treatment regimen as it alleviates the patient pill burden [15,25,26,27].

### 3.2. Simplified Dosing Schedule

The most notable advantage of FDC therapy is the ease and convenience of medication administration in this single dosage form. Additionally, FDC therapy is associated with improved safety based on simplification of the dosing routine, which reduces the potential for confusion when administering the medications [25]. The easing of complicated dosing regimens ultimately improves patient compliance due to a reduced pill burden as shown in the example of first-line TB treatment in Table 1 [26]. In the example presented below, two scenarios are presented for both the intensive- and maintenance treatment phases. These examples capture a ‘best’ and ‘worst’ case scenario based on product availability which can influence the number of tablets administered daily to achieve a therapeutic dose.

The concept of FDC therapy is always met with great argument as soon as it extends beyond a single-tablet regimen. The proposal by Tsiligiannis et al., 2019 to dose according to therapeutic exposure levels requires a regimen ranging from one to four mini tablets administered daily [29]. This calls into question the usefulness of such an FDC product if a patient is required to take four of the same tablets daily as opposed to four tablets comprising the individual drugs, as the sum of tablets administered per day for both options is equal. Whilst the number of tablets to be taken remains the same in such instances, advocates of FDC will point out that the pill burden to patients has been reduced in the sense that understanding the dosing instructions has been simplified from four counts to just one [25,27]. Increased safety associated with the simplification of a therapeutic regimen relates to decreasing the potential for confusion surrounding dose administration and intervals [25].

### 3.3. Prevention of Dose Dividing

FDC therapy has the innate ability to prevent dose dividing or sharing of medication. Dose dividing refers to when a patient takes less than the indicated quantity of their daily medication to make their medication supply last longer or, alternatively, medication is shared with family members exhibiting similar symptoms. Both instances are driven by the cost of medication as well as access to and availability of medication [30]. FDC limits the chance of a patient receiving less than the daily therapeutic dose, as the whole dose is incorporated into one tablet as opposed to multiple tablets per treatment regimen.

FDC formulations are strongly preferred over the co-packing of several drug products for the treatment of a single condition. Co-packing refers to a pharmaceutical product consisting of two or more separate dosage forms (tablets or capsules) presented in their final dosage form that are packaged together for distribution to patients. FDC therapy mitigates the risk of patients incorrectly administering co-packaged products, sharing their medication with family members, or trying to extend their supply of medication by only administering one of the co-packaged products daily as opposed to the prescribed two-drug regimen. The use of individual components intended to be administered concomitantly leads to sub-therapeutic drug levels leading to disease mismanagement and emergence of drug resistance. Thus, FDC therapy minimizes the probability of patients splitting the prescribed dose or taking only some of the drugs in the regimen.

### 3.4. Drug Resistance Prevention

FDC therapy prevents dose diving, which is an established problem with individual therapeutic components that are co-packaged or individually packaged. Furthermore, simplification of dosing regimens minimizes the opportunity for confusion and the incorrect administration of medication leading to sub-therapeutic or toxic drug levels. The use of the individual therapeutic components intended to be used concomitantly leads to the emergence of drug resistance, as no crossover protection is provided [31]. Fixed-dose combination therapy can play an important role in the effective treatment of infectious diseases such as HIV, AIDS, TB, and malaria, which have exhibited enormously high morbidity and mortality rates to date, and whose treatment continues to be hindered by the emergence of resistance to available therapeutic regimens. The research and development of new therapeutic agents is a costly and time-consuming process with many pipeline drugs failing to progress through the phases of clinical trials due to dose-limiting toxicities or sub-therapeutic levels. Therefore, the WHO recommends the use of combination therapy, preferably formulated as FDC products, to provide crossover protection and limit the emergence of drug resistance. Whilst polytherapy administered as individual therapeutic components provides crossover protection, there is an increased risk of the incorrect administration of the drug products, whether it be due to patient noncompliance resulting from high pill burden, confusion relating to complicated dosing regiments, a socioeconomic need to ration medicine, or a combination of these factors. Therefore, FDC formulations ensure that multiple drugs are administered simultaneously to provide cross-over protection. Examples of therapeutics that employ crossover protection include antibiotics, anti-malarial (Section 5.2), anti-tuberculosis (Section 5.3), and antiretroviral (Section 5.4) FDC products.

### 3.5. Synergism

Synergism occurs when two drugs act at different sites and one drug increases the efficacy of the other drug by either changing the biotransformation, distribution, or excretion [32]. Further, synergistic drug interaction can refer to one co-administered drug correcting the side effect(s) associated with the other drug, such as the concurrent administration of a diuretic to correct the salt and water retention occasionally associated with β-blocker therapy [33]. To that end, calcium-channel blockers and renin-angiotensin system blockers in combination demonstrate additive protective effects on the vascular wall [34]. 

Using a combination of agents to target different steps within the HIV life cycle provides either a synergistic or additive antiviral effect, thus enhancing the efficiency of viral suppression [35]. Additionally, the rational combinations of synergistic agents in TB treatment, administered as an FDC to enhance patient compliance, aids the prevention and management of drug-resistant tuberculosis. Chlorpromazine in combination with the frontline agents, rifampicin and isoniazid, demonstrated clear synergy, thereby suggesting the capacity for this combination to restore drug activity against mutant strains genetically resistant to either of the partner compounds. This was further demonstrated by employing a combination comprising rifampicin, spectinomycin, and chlorpromazine, which was found to enhance in vitro activity against *M. tuberculosis*. This three-drug combination is active in *M. tuberculosis*-infected macrophages and against mono-resistant, pre-MDR strains [36]. The advantages of FDC therapy are summarized in Figure 1.

## 4. Challenges of Fixed-Dose Combination Formulations in Drug Therapy

### 4.1. Time-Dependent Patient Compliance

Adherence to polypill regimens is significantly higher compared to multiple-pill regimens [39]. However, regardless of the number of tablets administered, there is a time-dependent decrease in compliance associated with treatment regimens extending beyond 18 months of therapy [24,40]. This suggests that whilst FDC therapy can improve patient compliance through easing the pill burden, ultimately, the nature of a chronic condition, i.e., the duration of the treatment period, is the greatest hindrance to patient compliance, more than the type of dosage form. With that said, it remains of utmost importance to design treatment regimens with patient convenience, and therefore compliance, in mind, to achieve the best therapeutic outcomes over an extended treatment period.

Interestingly, the Kanyini-GAP trial looked at the necessity and prevalence of prescribing other drugs in addition to the two polypills investigated in the trial once the predetermined follow-up period of 18 months was reached [25,41]. Although the prescription of an additional drug to use in conjunction with the already prescribed polypill would appear to negate the overall premise of an FDC therapy regimen (i.e., the simplification of dosing regimens and reduction of pill burden), the study found the overall pill burden to be reduced by means of two drugs, and participants recorded an overall outcome of 49% improvement in adherence [25,41].

### 4.2. Dose Inflexibility

The ability to titrate an individual component to reach desired therapeutic outcomes is the primary challenge in FDC therapy in certain disease states, such as CVD and diabetes mellitus [37]. This is less of a concern in conditions such as malaria and TB but is a potential concern when prescribing anti-retroviral therapy based on patient weight, which can result in patients having to break tablets in half to achieve the prescribed dose in FDC products that are not available in the required strength. Similarly, adjustments to individual components to minimize side effects or toxicity are not always possible should the adjusted dosing ratio not be available in an FDC product and are among the primary reasons why many prescribing physicians have been reluctant to fully embrace the use of FDC therapy [16]. Figure 2 describes a pentagonal framework of identified potential barriers opposing FDC therapy usage. Dosage form size restrictions are elaborated upon further below.

### 4.3. Development of Analytical Methods

Post-formulation, simultaneous determination of multiple APIs can require complex analytical techniques or the development of new analytical methods, which in itself can be a costly and time-consuming process requiring highly specialized equipment and expertise [42]. Whilst each individual component incorporated into the FDC formulation will have its own analytical method, this does not mean that that analytical method will be appropriate for the quantification of all components incorporated into the newly formulated FDC product. For example, the FDC product components may have differing high-performance liquid chromatography (HPLC) requirements (solvent, pH, photosensitivity). Whilst more specialized techniques, such as liquid chromatography mass spectrometry (LC-MS), may be employed to simultaneously identify the various compounds, there are limitations on the machine pump based on the solvents used during analysis. Furthermore, the individual sample run time needs to be considered, as the solvent gradient may need to be adjusted during the run time to accommodate the different compounds. Alternatively, UV-spectrometry may be employed; however, not all compounds absorb ultraviolet light and, therefore, this method may not be suitable for the quantification of all analytes of the newly formulated FDC product. This is not to say that analytical methods for the simultaneous determination of FDC product components cannot be developed. This is merely a brief list of the type of considerations needed during simultaneous detection method development.

To further illustrate this point, the characteristics of tenofovir disoproxil fumarate (TDF) and lamivudine (3TC) additionally highlight the depth of review and consideration required. TDF possesses one chiral center translating to two stereoisomers with (R) as the pharmacologically active enantiomer and (S) as the undesired isomer. Similarly, 3TC has two chiral centers with four stereoisomers, of which (2R,5S)-3TC is the desired enantiomer, and three undesired isomers, namely, (2S,5R)-3TC, (2S,5S)-3TC, and (2R,5R)-3TC [43]. There are a limited number of reported methods for chiral purity estimation of 3TC or TDF individually [44,45,46,47]. Kurmi et al., 2020 have reported the first method for chiral purity estimation of both drugs simultaneously when incorporated in an FDC formulation. Development of a single method to determine chiral purity for the combination of 3TC and TDF is challenging due to the possibility of six stereoisomers in total within the mixture [43]. This example highlights how time-consuming the process of developing an FDC product can be, as the foundation of a new product extends beyond the selection of compatible components and the correct method of manufacture. 

### 4.4. Drug Interactions

Drug-drug and drug-excipient interactions are important considerations when formulating FDC products. Drug-drug interactions may influence bioavailability as is the case with rifampicin and isoniazid. Although rifampicin-isoniazid is a combination approved and recommended by the WHO, rifampicin has demonstrated instability in the presence of isoniazid when exposed to an acidic environment, which decreased the bioavailability of rifampicin [48,49,50,51]. Another example of drug-drug interactions may occur with antihypertensive drug use in patients with comorbidities and polytherapy. Calcium channel blockers, namely diltiazem and verapamil, are highly likely to induce significant drug interactions due to their potent inhibition of CYP-3A4 [52]. Anti-inflammatory drugs and steroids are known to affect blood pressure and should be avoided in combination with anti-hypertensive medications [53]. Moreover, thiazide diuretics and β-blockers in combination increase diabetogenic risk, thereby demonstrating the need for prescribers to be aware of comorbidities in their patients and the associated implications of certain combinations. Interestingly, when thiazide diuretics or β-blockers are used in combination with renin-angiotensin system blockers, the diabetogenic risk is reduced in comparison to those agents as monotherapy [54]. A comprehensive understanding of synergistic or antagonistic effects on blood pressure needs to be present during dosage form design and prescribing.

### 4.5. Fixed-Dose Combination Therapy Conflict with Personalized Medicine

There is a need to tailor treatment approaches to individual patient needs due to differences in genetic profiles, race, gender, age, epigenetic, as well as environmental factors. Patient weight, comorbidities, and inter-patient tolerance and side effects of therapy necessitate personalized treatment plans. A one-size-fits-all approach is a thing of the past in this regard and is thereby heavily undermined by FDC therapy that lacks dose flexibility [55]. However, the practicality and cost implications of personalized medication are a barrier to this therapeutic approach. Additive manufacturing has been proposed as a production method to manufacture personalized solid oral dosage forms. This technology is undoubtedly only suited to higher-income areas, and the highly specialized equipment is not yet readily accessible in developing countries [56,57]. Further shortcomings are the lack of specialized and approved pharmaceutical materials, which is covered in greater detail in a section below. Additive manufacturing is still in its infantile stages and requires considerable funding and input to develop this approach as a feasible product delivery option. The rationale for personalized medication is sound, and additive manufacturing is a plausible manufacturing method currently under investigation globally. In the meantime, until this becomes a credible, viable, and widely available manufacturing approach, established and conventional manufacturing techniques will remain the primary approach to formulate FDC therapies.

### 4.6. Individual Drug Patents Hinder Development of Fixed-Dose Combination Products

Fixed-dose combination products often incorporate APIs with expired patents and seldom include new molecular entities. The approval of FDC products can be delayed based on the patent status of each individual component intended for inclusion in the proposed FDC product. Pharmaceutical companies have been known to develop and market an FDC product comprising an API that has a patent nearing expiry. This strategy is intended to extend the patent and exclusivity life of the API. Efforts are being made to exclude FDC products from single API patent restrictions and instead create a medicines patent pool to ease access to the proposed FDC regimen. Most FDC products include at least one single API under patent, with only a small percentage of FDC products approved using the Food and Drug Administration (FDA) priority review procedure [58].

## 5. Conditions Commonly Treated with Fixed-Dose Combination Formulations

### 5.1. Cardiovascular Disease

Several antihypertensive FDC products have been added to the WHO model list of essential medications [59]. The use of FDC products has a proven record of being able to lower blood pressure in patients with hypertension to goal levels. This is seldom achievable with the use of single-drug therapy during stage I or II hypertension, even when maximally titrated. The success of multiple-drug treatment with FDC formulations in reducing blood pressure can be attributed to the incorporated APIs targeting different effector pathways. Additionally, one API may trigger the counter-regulatory system activity, which is kept in control by the other API in the FDC formulation [60]. This is demonstrated with the example of a diuretic and β-blocker combined in an FDC product. The diuretic will correct the salt and water retention occasionally associated with β-blocker therapy. In other words, the APIs in an FDC formulation not only treat the disease in question but can also counteract the adverse effects of their co-administered components [61]. This is further illustrated by the co-administration of an angiotensin-converting enzyme (ACE) inhibitor in hypertension drug therapy to mitigate the peripheral oedema that accompanies calcium channel antagonists due to the venodilation imposed by ACE inhibitors. Moreover, diuretic-induced volume contraction may generate a secondary hyperaldosteronism state leading to electrolyte abnormalities such as hypokalemia and/or hypomagnesaemia. In these instances, the co-administration of either an ACE inhibitor or an angiotensin II receptor blocker together with a diuretic will correct the aforementioned electrolyte disturbances [33]. Furthermore, the combination of these ingredients has an additive antihypertensive effect, reducing blood pressure to a greater degree than either component alone [62,63].

In the last decade, the development of multiple FDC formulations or polypills has been witnessed for the treatment of CVD, as documented in Table 2. The FDC products listed in Table 2 adhere strictly to the definition of a polypill for CVD prevention, i.e., formulations comprising at least one anti-hypertensive drug in addition to aspirin and a lipid-lowering medicine, namely a statin.

Table 3 documents a wide range of FDC products appropriate for use in CVD prevention but incorporating constituents that deviate from the original CVD polypill definition.

### 5.2. Antibacterials for Systemic Use

Rising antimicrobial resistance is a global health concern with serious consequences for morbidity and mortality that requires HCPs and patients to conscientiously enforce antibiotic stewardship. A noteworthy FDC product recommended by the WHO is that of co-trimoxazole. This combination of sulfamethoxazole and trimethoprim is prophylactically used to prevent infections in high-risk, HIV-positive patients. This FDC can improve treatment response through synergistic mechanisms of action [65]. Examples of antibacterial FDC products for systemic use are reported in Table 4.

There are, however, clinically inappropriate combinations of antibiotics that offer unnecessary broad-spectrum coverage which, when used incorrectly and frequently, contribute further to the emergence of antimicrobial resistance. For this reason, the combination of antibiotics in FDC products must have a sound rationale with full microbiological, pharmacological, and clinical validation and safety studies [66]. It is recommended that antibiotic-adjuvant FDC products should be investigated, as opposed to developing antibiotic-antibiotic FDC products [67].

### 5.3. Malaria

Artemisinin-based combination therapy is recommended by the WHO as standard of care in malaria and encompasses the use of an artemisinin derivative with a partner drug in either double or even triple FDCs (Table 5).

Artemether and lumefantrine are approved and recommended in a fixed-dose ratio of 1:6. The two incorporated drugs have different mechanisms of action and thereby different target sites as well as different elimination half-lives [8]. Lumefantrine is absorbed and eliminated relatively slowly [71], which renders it a suitable candidate in artemisinin-based combination therapy as it will eliminate any remaining parasites after the short-acting artemisinin derivative (such as artemether) has reduced the initial parasite burden and initial malarial symptoms [72]. From a resistance prevention perspective, the drugs used in combination should have similar elimination rates to provide optimum mutual protection against resistance [73]. However, there are benefits to one of the drugs having a slower elimination rate as this allows for a shorter therapeutic regimen to be followed by the patient, which could enhance patient compliance [74,75]. In this instance, the drug with the slower elimination rate (lumefantrine) provides protection against Plasmodium species after the final combination dose of artemether and lumefantrine has been administered. Lumefantrine has a half-life of 4.5 days, meaning there is some protection after the final dose is administered and the action of artemisinin has decreased [76,77]. Lumefantrine has an elimination half-life of 2–3 days in healthy volunteers and 4–6 days in malaria-infected patients, which provides some protection against Plasmodium species after the final dose was administered and the action of artemisinin has subsided [76,78]. A challenge facing this therapy relates to the elimination rates of the therapeutic agents available. Since the artemisinin derivatives in artemisinin-based FDC products are eliminated rapidly, and the partner drugs are eliminated slowly, there is complete protection for the artemisinin derivative only. The combination still provides good protection against the emergence of resistance to the partner drug; however, the risk of developing resistance is elevated in the drug with slower elimination [73].

Ganaplacide in combination with lumefantrine has been developed by Novartis as a formulation optimized for once-daily dosing with improved solubility and oral bioavailability. This formulation is currently under Phase III investigation (NCT03167242) and aims to reduce the risk of resistance and maintain efficacy. Should this combination be approved, it will be the first non-ACT treatment since Malarone^®^ (atovaquone-proguanil) launched in 2000 and it would provide significant cover in the event of artemisinin resistance emerging [79,80].

### 5.4. Tuberculosis

The WHO and the International Union Against Tuberculosis and Lung Disease (IUATLD) recommend rifampicin and isoniazid in an FDC as first-line therapy to ensure optimal treatment of TB and minimize the emergence of resistance [10,81]. The current WHO-approved first-line treatment regimen for TB in adults above 55 kg body weight encompasses a four-drug combination of rifampicin (150 mg), isoniazid (75 mg), pyrazinamide (400 mg), and ethambutol hydrochloride (275 mg) for the first two months of treatment, known as the intensive phase [81]. Thereafter, rifampicin (300 mg) and isoniazid (150 mg) are co-administered for a further period of four months (continuation phase), leading to a total treatment period of six months. Examples of FDC products for use in the first-line management of TB treatment are listed in Table 6.

The WHO has recommended the use of FDC therapy over separate drug dosing for TB treatment from a drug resistance perspective due to its efficiency in the reduction of viable bacteria together with high patient satisfaction [7,10]. The evidence presented in works by Albanna et al., (2013) and Gallardo et al. (2016) demonstrated rifampicin-isoniazid FDC therapy to be non-inferior and as effective as separate drug formulations regarding treatment failure, adverse events, and death [5,6,10]. Most importantly, this FDC therapy has yet to be investigated regarding bioavailability and, therefore, comparisons with separate drug formulations cannot be made. Limitations on the total size/weight of a single solid oral dosage form pose the greatest challenge when incorporating multiple drugs into a single oral dosage form for administration. However, a product such as Rifafour^®^ e-275 (Sanofi-Aventis) has proven it is possible to incorporate multiple, high-dose drugs into an FDC product comprising rifampicin (150 mg), isoniazid (75 mg), pyrazinamide (400 mg), and ethambutol (275 mg) into a single tablet. Great care needs to be taken when incorporating multiple drugs into a single product to ensure that the drugs remain stable during manufacturing, the shelf life, and after administration.

The characteristics of rifampicin remain the primary challenge surrounding TB treatment. Rifampicin quickly develops drug resistance when used in isolation, thereby warranting the need for multiple drug therapy that allows crossover protection offered by the partner drug(s). However, rifampicin has demonstrated instability in the presence of isoniazid when exposed to an acidic environment, ultimately impairing the bioavailability of rifampicin [48,49,50,51]. Moreover, its behavior as a zwitter-ion and its varied bioavailability cause sub-therapeutic levels, which in turn contributes to the emergence of drug resistance and treatment failure [50,82]. Studies investigating the effect of co-administration of ascorbic acid have found that it slows down the degradation of rifampicin in the acidic environment [83]. The effects of ascorbic acid on the stability of rifampicin in an FDC formulation were first investigated by Subashini et al. (2017), and there is a need for additional studies to be conducted in order to supplement this work to optimize TB therapeutic agents [7].

The relationship between possible individual patient variables and the pharmacokinetic parameters determining the exposure levels deemed to be therapeutic needs to be considered [84,85]. Tsiligiannis et al., 2019 investigated the optimal dosing strengths of FDC mini-tablets comprising rifampicin, isoniazid, and pyrazinamide 95/75/200 mg, proposed for four body weight bands denoted as 4–8 kg, 8–12 kg, 12–18 kg, and 18–28 kg [29]. Children with a body weight ≥28 kg were treated with adult doses. Research such as this highlights a pediatric-focused approach to formulating, namely flexible dosing regimens, and reduces patient pill burden through simplification of the dosing regimen, i.e., pediatric patients receive between one and four mini-tablets, depending on their weight.

In summary, the provision of medication to children is more complex than the provision to adults due to challenges such as drug doses, which need to be reviewed regularly to keep up with child growth; the palatability of formulations is generally poor; and caregivers often have practical difficulties in dispensing liquid formulations, as drawing up liquids requires good vision and basic arithmetic skills [86]. There is a need for specialized pediatric formulations that simplify dosing regimens, are more palatable, and possess an appropriate formulation composition for children.

### 5.5. Human Immunodeficiency Virus

Long-term adherence to antiretroviral therapy among patients living with HIV is critical for achieving virologic suppression, reducing the risk of transmission to uninfected partners, and minimizing the risk of drug resistance developing. The complexity of treatment regimens, characterized by multiple tablets, differing dosing times, as well as varying dosing instructions all contribute to suboptimal adherence [87]. Examples of FDC antiretroviral therapeutic options are listed in Table 7.

The greatest unmet need in the HIV pediatric population is the availability of suitable formulation options. Primarily anti-retroviral therapy dosage forms include tablets and capsules intended for adults, which are inappropriate for the pediatric population. Historically, suspensions are extemporaneously prepared by crushing adult-sized tablets, which were unsafe for administration to children due to the risk of choking. It is only in recent years that pediatric FDC tablet preparations have been introduced (Table 8). The primary challenge surrounding the pediatric population is the need for flexible dosage regimens based on variations in patient weight [88,89]. Dose adjustments and increments are based on individual patient body weight, which differs significantly among children and traditionally translates to a milligram/kilogram-based dose [90]. This dosing approach proves problematic, as the relationship between drug clearance and body weight is allometric in children [90,91,92,93], and can thus lead to under- or overdosing [94,95,96,97]. The intricacy of dosing is further complicated by inter- and intra-patient variables, such as stages of development, maturation of liver enzymes, and comorbidities that will affect the pharmacokinetics in young children [91,98].
pharmaceutics-16-00178-t007_Table 7Table 7Examples of antiretroviral fixed-dose combination products formulated in a single-tablet regimen for adults and adolescents [28,35,99].Brand Name(s) and CompaniesActive Pharmaceutical Ingredients ClassificationNRTIs/NtRTIsNNRTIsINSTIPIPK EnhancerAluvia^TM^ (Abbott Laboratories); Kaletra^®^ (AbbVie)


Lopinavir (100; 200 mg)Ritonavir (25; 50 mg)Atripla^®^ (MSD); Atroiza (Mylan); Citenvir (Novagen); Odimune (Cipla); Tribuss™ (Aspen)Emtricitabine (200 mg)Tenofovir disoproxil (300 mg)Efavirenz (600 mg)


Biktarvy^®^ (Aspen)Emtricitabine (200 mg)Tenofovir alafenamide (25 mg)
Bictegravir (50 mg)

Combivir^®^ (GlaxoSmithKline, ViiV); Combozil (HeteroDrugs SA); Duovir (Cipla); Adco-lamivudine and zidovudine; Lamzid (Aspen); Loziv (Novagen Pharma)Lamivudine (150 mg)Zidovudine (300 mg)



Complera^®^ (Aspen), Eviplera^®^ (Gilead)Emtricitabine (200 mg)Tenofovir disoproxil (300 mg)Rilpivirine (27.5 mg)


Delstrigo^®^ (Merck & Co.)Lamivudine (300 mg)Tenofovir disoproxil (300 mg)Doravirine (100 mg)


Dovato (GlaxoSmithKline)Lamivudine (300 mg)
Dolutegravir (50 mg)

Epzicom^®^ (US, ViiV), Kivexa^®^ (GSK, ViiV Healthcare); Dumiva (Mylan)Abacavir (600 mg)Lamivudine (300 mg)



Duovir-N (Cipla); Triomune (Cipla); Sonke LamNevStav; Virtrium (Aspen)Lamivudine (150 mg)Stavudine (30 mg)Nevirapine (200 mg)


Genvoya^®^ (Aspen)Emtricitabine (200 mg)Tenofovir alafenamide (10 mg)
Elvitegravir (150 mg)
Cobicistat (150 mg)Juluca (GlaxoSmithKline)
Rilpivirine (25 mg)Dolutegravir (50 mg)

Odefsey^®^ (Aspen)Emtricitabine (200 mg)Tenofovir alafenamide (25 mg)Rilpivirine (25 mg)


Stribild^®^ (Gilead)Emtricitabine (200 mg)Tenofovir disoproxil (300 mg)
Elvitegravir (150 mg)
Cobicistat (150 mg)Symfi^®^ (Mylan); Tenarenz (Aspen)Lamivudine (300 mg)Tenofovir disoproxil (300 mg)Efavirenz (600 mg)


Symtuza^®^ (Janssen Pharmaceutica)Emtricitabine (200 mg)Tenofovir alafenamide (11.2 mg)

Darunavir (800 mg)Cobicistat (150 mg)Triplavar (Cipla)Lamivudine (150 mg)Zidovudine (300 mg)Nevirapine (200 mg)


Triumeq^®^ (GlaxoSmithKline)Abacavir (600 mg)Lamivudine (300 mg)
Dolutegravir (50 mg)

Trizivir^®^ (GlaxoSmithKline)Abacavir (300 mg)Lamivudine (150 mg)Zidovudine (300 mg)



Truvada^®^ (Gilead); Adco-Emtevir (Adcock Ingrams); Tencitab (Aspen); Didivir (Cipla); Tyricten (Aurobindo); Tenemine (Mylan)Emtricitabine (200 mg)Tenofovir disoproxil (300 mg)



INSTI, Integrase strand transfer inhibitors; NRTIs/NtRTIs, nucleoside and nucleotide reverse transcriptase inhibitors; NNRTIs, non-nucleoside reverse transcriptase inhibitors; PI, protease inhibitor; PK, pharmacokinetic.


### 5.6. Diabetes Mellitus

Most patients with type 2 diabetes mellitus do not achieve the recommended glycemic levels with monotherapy and often require multiple anti-hyperglycemic agents to achieve glycemic control [38]. The biguanide metformin is generally considered as the first-choice medication. This is predominantly due to the anti-hypoglycemic efficacy of metformin coupled with its favorable effect on body weight, and low cost [101]. Should metformin monotherapy fail to attain sufficient glycemic control, treatment guidelines recommend the addition of complimentary pharmacotherapeutic agent(s) as listed in Table 9 [101]. FDC therapies have been found to offer greater efficacy compared with higher-dose monotherapy, reduce the risk of adverse reactions in comparison to higher-dose monotherapy, and offer improved medication concordance [38,102].

Wang et al. 2013 reported that acarbose-metformin in FDC formulations yielded superior anti-hypoglycemic efficacy, with proportionally more diabetes mellitus type 2 patients reaching HbA1c targets with reduced overall body weight when compared to acarbose monotherapy [103]. The researchers further reported that the acarbose-metformin combination was well tolerated amongst their study population and devoid of hypoglycemia risks [103]. Such results, positively reporting on FDC therapy, need to be reinforced with more robust studies so that definitive conclusions of therapeutic benefits can be drawn. Whilst multiple studies have reported the efficacy and safety of acarbose as add-on therapy in situations where metformin ineffectively controlled glycemic levels, limited results are available reporting on the efficacy of metformin as a second-line add-onto to acarbose [104,105,106,107]. This speaks to an overall trend that requires a more in-depth, comprehensive evaluation of FDC products to better understand the efficacy of proposed combinations.

The most common chronic complication of diabetes mellitus is termed peripheral neuropathy, which, due to its high prevalence of up to 50% in diabetic patients, is synonymously called diabetic neuropathy [108,109]. This condition arises secondary to pre-diabetes, type 1 diabetes mellitus, or type 2 diabetes mellitus, which is a result of the persistent high glucose levels of these patients [110]. Pre-diabetes exhibits a similar pattern of nervous system damage, thereby supporting the cause of nerve injury as the continuum of fluctuations from normal glycaemia [108]. A definitive treatment for peripheral neuropathy remains elusive, and thus preventing disease progression and further complications remains the principal approach to reducing the severity of this condition. Management of peripheral neuropathy should also include effective treatment of pain. Furthermore, inflammation plays a critical role in the regulatory processes involved in the degeneration of nerves in patients with peripheral neuropathy, and thus the combination of selected B-vitamins (i.e., vitamins B_1_, B_6_, and B_12_) as adjunct therapy to diclofenac has been investigated in the clinical management of peripheral neuropathy. Current prescribing practices based on market availability require separate administration of these products, leading to high pill burden among these patients. The development of an FDC formulation containing diclofenac and vitamins B_1_, B_6_ and B_12_ into a single-tablet regimen requires further investigation, as it has the potential to improve therapeutic outcomes.

## 6. Fixed-Dose Combination Formulation Factors for Consideration

The importance of dosage form design for efficient drug treatment cannot be overstated. A successful dosage form must ensure predictable and repeatable delivery of the API(s) to reach the site of action at the intended concentration [111]. The design of FDC dosage forms is a complex task requiring multiple simultaneous considerations [112]. When considering the formulation of an FDC product, the reality remains that the difficulty of manufacturing the dosage form increases with the inclusion of each additional API. From a clinical perspective, the combination of multiple APIs increases the potential risk for patients to develop side effects. From a technical standpoint, formulation complexity correlates with the number of APIs incorporated into an FDC product due to the chemical properties and inherent physical attributes of each API.

The formulation scientist needs to research each moiety and consider the chemical and physical stability of the components when in contact with each other. Thereafter, methods of manufacture should be taken into consideration. For example, if one of the APIs is sensitive to moisture, all manufacturing processes utilizing water are ruled out. Factors including thermolability, sensitivity to compression force, and stability throughout the gastrointestinal tract (GIT) influence the selection of the manufacturing method. Moreover, additional process steps such as coating may be necessary as a physical barrier for acid-labile APIs or drugs known to cause gastric irritation. Layered tablets with modified-release properties, a mixture of drug-containing granules, or tablet-in-capsule systems can be employed to achieve physical separation of the APIs. Additionally, physical separation of the components post-administration may be achieved by formulating matrix systems or reservoir systems with modified- and delayed-release properties. Alternatively, novel formulation techniques such as additive manufacturing may be employed [113,114]. Figure 3 provides a simplified schematic representation of the factors for consideration during the FDC dosage form design process.

The dosage size of the respective APIs for inclusion in the FDC dosage form may be a highly restrictive factor, as maximum tablet sizes are approximately equivalent to 1000 mg or capsule equivalent to size 000. The selected APIs in combination with excipients can result in a dosage form being an inappropriately sized solid oral dosage form (i.e., physically too large for oral administration). The feasibility of obtaining an appropriately sized tablet is determined by the compressibility of the APIs and selected excipients when employing direct compression as a means of manufacture. The SeDeM Expert Diagram System may be very useful to a formulation scientist as a pre-formulation aid, as it considers 12 parameters relating to powder properties (such as bulk- and tapped density; inter-particle porosity; Carr’s index; cohesion index; Hausner ratio; angle of repose; etc.) to determine the suitability of a powder for direct compression as well as to predict the drug loading capacity of investigated formulations [116,117,118,119,120].

## 7. Formulation Approaches

Formulation approaches relevant to FDC tablets are discussed with examples of commercially available FDC tablet products. The goal when formulating solid oral dosage forms is the same for FDC tablets as it is for monotherapy tablets, i.e., to establish a mass-production setup with a robust and quality-controlled approach resulting in acceptable and elegant looking preparations of consistent quality (e.g., uniform product weight and dose) [121].

### 7.1. Immediate-Release Fixed-Dose Combination Dosage Forms

Immediate-release dosage forms intend to rapidly release the APIs after administration to achieve a fast onset of action, following absorption in the GIT [111]. Immediate-release tablets are the most common type of tablet, synonymously termed conventional tablets, and are considered the standard and easiest approach to deliver APIs for a rapid therapeutic effect [122]. The preparation procedure commonly employed is that of direct compression. This is a simple process in which APIs are blended with the excipients prior to tableting, and it consists of fewer preparation steps than other methods such as wet granulation [123]. Direct compression is well suited to thermolabile and hydrolabile drugs, as no heat or water is required during the process steps. One challenge associated with direct compression is that the compression mixture needs to be able to flow effectively through the hopper into the tablet die to ensure tablets of consistent weight, thereby delivering accurate dosing of the drug. Therefore, the flowability and compressibility of the excipients together with their quality and consistency play a vital role in the success of directly compressed FDC tablets when used as a production aid to supplement the poor powder flow of the APIs [124]. Important considerations when utilizing this method of preparation include differences in particle sizes of the APIs themselves as well as the selected excipients for inclusion in FDC tablets. Significant differences in particle size can result in uneven distribution of the components within the physical mixture. Should variations in particle size result in a heterogenous mixture, steps such as sieving for a select particle size range should be employed or granulation methods should be considered in which the particle size can be controlled [125,126].

Physical and chemical interactions between the APIs both pre- and post-administration need to be ruled out, as the APIs will be in continual contact once tableted and could further interact within the acidic environment of the GIT. Further analysis should be conducted to establish that no incompatibilities exist between the APIs and selected excipients. For instance, stearate lubricants should be avoided in combination with ibuprofen [127]. Extensive consultation of the literature as well as tests such as differential scanning calorimetry, thermogravimetric analysis, x-ray powder diffraction, and thermal activity monitoring can be employed to determine mixture stability [128].

Examples of immediate-release FDC tablet products include amoxicillin in combination with clavulanic acid as well as many over-the-counter pain preparations which include combinations of aspirin, paracetamol, and caffeine, or paracetamol, codeine, and ibuprofen.

### 7.2. Modified-Release Fixed-Dose Combination Dosage Forms

Modified drug release refers to the manipulation or modification of the drug release profile from a dosage form with the specific intention to deliver the API(s) to a specific absorption site within the GIT at a desired rate or at a pre-determined time point [111]. This is achieved through a variety of formulation techniques and excipients. Modified drug release can further be subdivided into extended-release, delayed-release, or gastro-resistant dosage forms, based primarily on the techniques employed.

#### 7.2.1. Extended-Release Dosage Forms

These dosage forms are primarily employed to allow for a reduction in dosing frequency, as it allows for the drug plasma levels to be sustained over a longer period of time thereby reducing the number of administrations required to maintain the drug within the therapeutic concentration. Extended-release systems can be achieved via matrix tablets, polymer-containing pellets, coated pellets or tablets, or osmotic-based systems [111].

##### Matrix-Type Drug Delivery Systems

The matrix drug delivery system refers to a type of extended-release dosage form that encompasses the dispersion of solid drug particles in a porous solid medium (i.e., the matrix) capable of prolonging drug release over an extended period of time. Typically, this is achieved by compressing a mixture of the APIs with a release-modifying polymer excipient into a matrix-type tablet [129]. Additionally, multiple-unit matrix systems may also be manufactured utilizing extrusion-spheronization, spray congealing, and casting. Drug release from matrix systems can be categorized as diffusion-controlled release in which the hydrophilic core remains intact and, over time, the swellable, soluble matrix hydrates, allowing the drug to dissolve and diffuse through pores in the system. Conversely, a hydrophobic polymer is selected to form the base of the matrix through which drug particles can leach out via hydrated channels within the matrix. The rate of drug release from these systems is dictated by the pore size, the number of pores, as well as the tortuosity of the matrix. Alternatively, erosion-controlled release can be employed, whereby the polymer and drug are continuously liberated from the surface of the matrix system through abrasion [122]. Polymers such as hydroxypropyl methylcellulose and polyethylene oxide as well as some natural polysaccharides have previously been employed to maximize the efficiency of their self-assembling properties to spontaneously form gel networks without the use of harsh reaction conditions and solvents. With that said, some natural polysaccharides are highly soluble in water, which can greatly impede their potential for use as release-modifying excipients in matrix-type drug delivery systems, and should therefore be carefully researched before considered for inclusion [130,131].

##### Multiple-Unit Pellet Systems

Pellets make up multiple-unit drug delivery systems which consist of small discrete sub-units, each containing a portion of the dose. The small units are typically administered by loading the sub-units into a sachet or capsule, or compressing them to form tablets. The chief characteristics leading to the popularity of multi-particulate systems include the decreased variability in drug release profiles, reduced intra- and interpatient variations in gastric transit time, and minimized dependability on the fed and fasted states [132,133,134]. Additionally, fast gastric emptying is achieved due to the small sub-units distributing more evenly throughout the GIT, leading to fewer side effects such as local gastric irritation and reduced susceptibility to dose dumping—even in dose failure [134,135]. In comparison to single-unit dosing, these multi-particulate systems provide more reproducible pharmacokinetic behavior, including more predictable plasma level concentrations and bioavailability. This is attributed to the individual sub-units being able to leave the stomach continuously even if the pylorus is closed, due to the individual particle size being smaller than 2 mm [136,137]. Whilst this approach boasts a multitude of advantages, one drawback is the reduced mechanical strength of tablets comprising the compressed pellet sub-units in comparison to conventional tablets, thus potentially requiring the addition of pharmaceutical excipients to overcome this challenge [138,139,140,141,142]. This creates challenges, as powder pharmaceutical excipients may segregate from manufactured pellets, given the particle size difference [143,144], and may further be complicated by restrictions of the maximum tablet or capsule size.

Multi-unit pellet systems (MUPS) present as popular multi-particulate candidates given the physical and mechanical properties imparted by the manufacturing process. MUPS can be prepared by means of extrusion-spheronization, which delivers consistent spherical particles and unique deformation behavior [145,146,147]. MUPS can furthermore be administered as either uncoated drug-containing pellets or functionally coated pellets, capable of offering modified drug release profiles [135,145]. Lopimune 40/10 (Cipla) is an example of an FDC pellet formulation comprising lopinavir (40 mg) and ritonavir (10 mg), specialized for the HIV-positive pediatric population [148].

Unlike conventional tablets made directly from powder particles, dust formation is reduced during the tableting of MUPS due to the agglomeration of fine powder particles during the wet mass extrusion process step. Manufactured pellets demonstrate superior flow properties in comparison to their individual counterparts, which allows for better flow from the hopper to the tablet die during compression, thereby increasing content uniformity due to less mass variation [137,149]. Whilst better powder flow results in higher processing speeds, such compositions also demand lower lubricant concentrations, thereby lowering processing costs [144,150]. This technique is well suited to formulations where differences in particle size of the APIs and excipients are a concern, as pellets negate concerns of segregation within a mixture.

The primary drawback of this method is, however, incompatibilities between the APIs and the wetting agent as well as poor compaction performance of the pellets [151]. Moreover, the APIs need to be able to withstand the drying phase of this process, whether it be elevated temperatures during oven drying or the harsh temperature reduction of freeze-drying. Therefore, when considering this as a method of manufacture, the combination of APIs for inclusion in FDC tablets needs to be screened for sensitivity to moisture, heat, and compression (during extrusion as well as tableting).

#### 7.2.2. Coating

Coatings for drug delivery systems are being researched continuously, as they can change the external properties of dosage forms without interfering with the internal structure. Coatings can be employed as protection against chemical and physical degradation of the drug or delivery system, alteration of the drug release profile, improvement of the appearance and organoleptic properties, as well as other benefits such as bio-adhesion or responsiveness to external stimuli [152].

Film coating can be used to mask poor taste, improve physical appearance, protect against moisture, or prevent brittle FDC tablets from chipping/cracking [153]. Commercially available FDC tablets such as Malanil™ (GlaxoSmithKline; atovaquone, proguanil hydrochloride 250/100 mg), Glucovance^®^ (Merck; glibenclamide, metformin 1.25/500; 2.5/500; 5/500 mg), Exforge^®^ (Novartis; valsartan, amlodipine 160/5; 320/5; 160/10; 320/10 mg), Co Exforge^®^ (Novartis; losartan, amlodipine, hydrochlorothiazide 160/5/212.5/; 160/10/12.5; 160/5/25; 160/10/25 mg), and Tribuss^™^ (Aspen; emtricitabine, tenofovir disoproxil, efavirenz 200/300/600 mg) are examples of film-coated FDC tablets.

Functional coatings can also be applied to alter the release of an API from formulated FDC tablets. For example, the use of enteric coating may be used to delay the release of diclofenac to prevent gastric irritation. Polymers such as Eudragit^®^ L100 and Eudragit^®^ S100 could be used to achieve the site-specific release of diclofenac in the duodenum, as these agents dissolve at pH 6.0 and 7.0, respectively. Fixed-dose combination tablet coating should be carried out after complete viscoelastic strain recovery of the FDC tablets, i.e., after 48 h has elapsed [154]. The major challenge surrounding the coating of pellets intended for inclusion in FDC tablets manufactured by means of direct compression is the compression-induced damage to the functional coating, which leads to diminished modified-release capacity and subsequent failure in dosage form design as well as loss of taste-masking or drug-stabilizing properties [145,155].

#### 7.2.3. Delayed-Release Dosage Forms

These dosage forms are formulated to release the APIs once a lag time has elapsed after administration. The motive for targeting the release of a drug to a specific absorption site within the GIT is to improve the therapeutic efficacy and reduce the side effects of certain drugs. This is achieved as different regions in the GIT offer increased dissolution or absorption based on the specific physiology of the site as well as the characteristics of the drug such as its solubility profile and regional permeability [156,157]. A dosage form encounters many environmental changes during transit through the GIT, experiencing fluctuations in pH, enzyme activity, and intestinal flora between the different regions. Whilst these varying conditions can be quite harsh, these region-specific differences can be utilized as opportunities to induce drug release and improve bioavailability [158].

Release in the small intestine may be preferred, as the pH increases dramatically from pH 1.2 in the stomach to 6.8 and 7.4 in the duodenum and jejunum, respectively [159]. This may be sufficient to prevent drug degradation of acid-labile drugs in the stomach, achievable via matrix systems, floating systems, or the use of functional coatings. Additionally, lipid-based formulations work on the premise that the presence of exogenous lipids prompts the release of bile, which helps to form mixed micelles into which lipophilic drugs may partition and solubilize, thereby augmenting their solubility and potentially their bioavailability [160,161]. Products such as Coartem^®^ or Riamet^®^ (Novartis; artemether, lumefantrine 20/120 mg) utilize this strategy.

#### 7.2.4. pH-Responsive Drug Delivery Systems

The use of pH-sensitive drug delivery systems is specifically applicable for drug delivery in the GIT where the pH varies substantially between the different regions. This variation in pH provides the opportunity to formulate responsive delivery systems that can optimize drug absorption [162]. Gastro-resistant systems can be employed to protect drugs against degradation in certain regions of the GIT as well as reduce irritation caused by high amounts of free drugs [162]. Arthrotec^®^ (Pfizer; misoprostol, diclofenac 200 µg/50 mg) is an example of an FDC tablet with a gastro-resistant core containing diclofenac sodium surrounded by an outer mantle containing misoprostol [163].

### 7.3. Layered Tablets

Multi-layered tablets are presented as one approach to incorporating multiple drugs into a single dosage form while separating them from each other [164,165,166,167]. Multi-layered tablets are well suited to situations where the combination of multiple drugs in one dosage form requires the controlled release of each substance. Multilayer tablets, however, possess unique production challenges, such as dose accuracy of each layer, cross-contamination between layers, physical interactions between constituents of each layer, as well as capping or delamination during coating or storage [168,169,170]. Examples of FDC layered tablets incorporating the antidiabetic drug glimepiride and metformin are Glimser-1^®^ (Alembic), Glycomett^®^-GP 0.5 (USV), GLIMKAR-M2-FORTE (Care Formulation Labs), GLYSAP-M1 (Elxir), and GLITIP-2 Tablets (IVA Healthcare).

The work of Sonvico et al. 2009 demonstrated the ability to achieve delayed drug release without the use of coating through a carefully designed matrix system. They investigated the development of a multi-kinetics FDC of gabapentin and flurbiprofen, formulated as multilayered tablets for oral administration [169]. Multi-layered tablets with three different release kinetics were formulated in one dosage form, targeting two delivery sites. The layers were ordered as follows: top layer, a floating hydrophilic matrix for gabapentin intra-gastric release; middle layer, a disintegrating formulation for gabapentin immediate release; bottom layer, an uncoated matrix, swellable but insoluble in the gastric fluid, for delayed prolonged release of flurbiprofen in the intestinal environment [169]. This dosage form is the epitome of formulation design and understanding material science. Such dosage forms utilize different polymers for delaying and prolonging the release of APIs and demonstrate the feasibility of layered tablets for FDC therapy.

It is also possible to achieve immediate release of one API and delayed release of the other API with an FDC tablet formulation. This is demonstrated by the work of Chun et al., 2021 in which bilayer tablets were manufactured consisting of high-dose metformin in a sustained-release layer with low-dose dapagliflozin L-proline incorporated into an immediate-release layer. The optimized bilayer tablet showed similar in vitro and in vivo profiles to the reference drug, demonstrating bioequivalence of the test product and the control drug [171]. The same concept is demonstrated by Vimovo^®^ (Grünenthal; naproxen, esomeprazole 500/20 mg), which has been developed as a sequential-delivery tablet system comprising an outer immediate-release esomeprazole layer with an enteric coated naproxen core [172].

### 7.4. Lipid-Based Formulations

Lipid excipients can be utilized to enhance the solubilization and absorption of highly lipophilic APIs in the intestinal luminal fluid. By exploiting the body’s natural physiological response to the presence of ingested lipids, this method of manufacture can enhance the bioavailability of lipophilic APIs by increasing the saturation solubility within the microenvironment and is well suited to compounds susceptible to hydrolysis [173,174,175,176]. Lipid-based formulations range from spray congealing and spray drying to oil-based suspension or solutions, emulsions, or self-emulsifying drug delivery systems (micro- or nano range) for inclusion in capsules [177]. Additionally, solid lipid dispersions comprising highly lipophilic APIs may be prepared by means of hot fusion (applying heat to melt a polymeric material) or hot-melt extrusion (pressure is applied to the molten mass) to prepare modified-release drug delivery systems [178]. The elevated temperatures involved in hot melt extrusion and hot fusion are unsuitable for thermolabile drugs, and appropriate thermoplastic behavior is a prerequisite for the selected polymeric material [174].

Many anti-malaria and anti-tuberculosis drugs are poorly water-soluble; thus, the rationale exists for the development of an FDC dosage form suited to the delivery of these highly lipophilic moieties. Solid lipid dispersions incorporated into matrix tablets may control the diffusion speed of water via pores into the matrix due to the lipid coating, and augment the dissolution rate [173,179,180]. An example of this is Coartem^®^ (Novartis) and Riamet^®^ (Novartis), which are examples of lipid-matrix type tablets comprising artemether and lumefantrine, both lipophilic antimalarial drugs [181].

### 7.5. Additive Manufacturing

Three-dimensional printing technology is a form of additive manufacturing, which relies on computer-aided design software and offers extreme design flexibility, as dosage forms can be tailor-made to any size, composition, shape, and internal structure [152,182,183,184]. The FDA approved the first 3DP tablet in 2015, namely Spritam^®^ (Aprecia; levetiracetam 250 mg). Whilst there are currently no approved three-dimensional printed FDC tablets commercially available, this method of manufacture has potential for future use and may be able to overcome limitations associated with more conventional pharmaceutical manufacturing techniques.

Compared to conventional pharmaceutical manufacturing methods, three-dimensional printing can create complex personalized products of unique design and overcome some conventional pharmaceutical unit operation challenges, i.e., milling, mixing, granulation, and compression [55,185,186]. Additionally, it offers advantages such as less material consumption, with reduced production costs; the ability to incorporate poorly water-soluble drugs or drugs with narrow therapeutic indices; the ability to attain high drug-loading with precision, particularly relevant for potent drugs; and fast production rates possible due to fast operating systems [186,187,188,189,190,191].

Fused deposition modelling is a method often employed to produce three-dimensional printing tablets employing polymeric material with appropriate thermoplastic behavior [174,184,185,191,192,193]. This method of manufacture can produce complex geometric shapes and could theoretically be employed to produce dual-compartment capsules to orally administer the anti-TB drugs, rifampicin and isoniazid, which require physical separation, for example [182]. Alternatively, instead of printing geometric shapes to house the APIs, there is potential to include the APIs in the printing filament, thereby meaning the drug is directly printed into an appropriate shape for administration. This approach would eliminate the need for conventional tableting excipients such as fillers, binders, and disintegrants. Three-dimensional printing requires APIs and excipients to be thermostable (due to the high process temperatures involved) with low hygroscopicity, as water liberated during melting and extruding can produce air pockets within the mixture and compromise the integrity of the filament [194].

Three-dimensional printing also offers exciting research opportunities in material science. Eudragit^®^ E 100 (Evonik Healthcare), Eudragit^®^ L 100-55, Affinisol™ 100cP (Colorcon^®^), Affinisol™ 4M, and chitosan are candidate material requiring further investigation for its use in manufacturing pharmaceutical grade filaments.

A most notable point is the selection of the correct equipment to manage the high sheer volume of these complex mixtures for filament production. There are many technical parameters that need to be established and controlled, including temperature control and monitoring across the entire process, die size and shape, pulling tension of the filament, and post-manufacture cooling. Significant advances have been made in the accessibility, variety, and cost of three-dimensional printers themselves [185], and when this translates across to materials specialized for pharmaceutical application, personalized medication will benefit vastly.

### 7.6. Multiple-Unit Delivery Systems

Capsules may be utilized as a container-drug delivery system filled with powder or non-powder fillings such as tablets, capsules, and pellets. A tablet-in-capsule multiple unit system, for example, can be designed to incorporate mini-tablets with modified-drug release properties (and various lag times of release) within the hard gelatin capsule or, alternatively, a combination of fillings can be employed such as mini-tablets and pellets incorporated into a single capsule [195]. Moreover, a combination of solid and liquid fillings may be encapsulated. Capsule-in-capsule multiple-unit systems comprise a formulated capsule (either liquid- or dry-filled) which is housed in an outer liquid-filled capsule [196,197]. Potential solid-solid multiple-unit systems comprising FDC regimens include aspirin and atorvastatin with/without clopidogrel; rifampicin, isoniazid, and pyrazinamide; and efavirenz, emtricitabine and tenofovir. Conceptual examples of liquid-liquid multiple-unit systems include artesunate and amodiaquine or arterolane and piperaquine phosphate.

### 7.7. Layered Tablets with Drug-Free Layers

Layered tablets incorporating a drug-free, excipient-only layer in tablets allow tablets to be split to obtain exact smaller doses, if desired. This tablet design approach overcomes two of the biggest challenges associated with FDC products: dose inflexibility and the ability to titrate doses. Moreover, layered tablets can be manufactured to incorporate drug-containing layers, separated by a drug-free, excipient-only layer which allows the incorporation of physically incompatible APIs [198].

## 8. Conclusions

There are many benefits of FDC therapy, and the most notable benefit for the patient is the simplification of treatment regimens. The limitation in dose flexibility is noteworthy but can be overcome by only initiating FDC therapy for patients who are already stabilized on the individual therapeutic components. The reality of medication side effects is ever present regardless of the dosing strategy, and the difficulty in identifying the root cause of an adverse drug event is not a problem inherent to FDC therapy. FDC therapy offers potentially significant savings resulting from lower treatment failure rate, lower case-fatality ratios, decreased emergence of drug resistance, and thus less money needed for the development of new drugs. FDC therapy holds future promise to improve the management of disease through synergism and reduced adverse events whilst enhancing patient compliance. By reducing resistance emergence, FDC therapy extends the lifespan of existing therapeutic agents. There are various formulation approaches, drug delivery systems, and materials available to formulate FDC tablets, with several examples of commercially available products already on the market. Lipid-based formulations, three-dimensional printing, and layered tablets with drug-free layers have been identified as formulation approaches that hold promise as future solid oral-dosage form formulation strategies. Functional excipients may be exploited to impart specialized properties or characteristics to a delivery system. Further, unmet needs were identified where exciting research opportunities exist such as pediatric-specific dosage forms for HIV-positive patients; robust studies examining interactions of FDC products to better understand their interaction and synergism as highlighted by acarbose and metformin; the additive effects of employing an FDC of diclofenac sodium with B-vitamins as adjuvant therapy; and the influence of ascorbic acid to limit degradation of rifampicin in the stomach. Furthermore, the use of formulation techniques such as layered tablets, lipid-based formulations, coatings, or three-dimensional printing to create a physical barrier between rifampicin and isoniazid should be investigated.

## Figures and Tables

**Figure 1 pharmaceutics-16-00178-f001:**
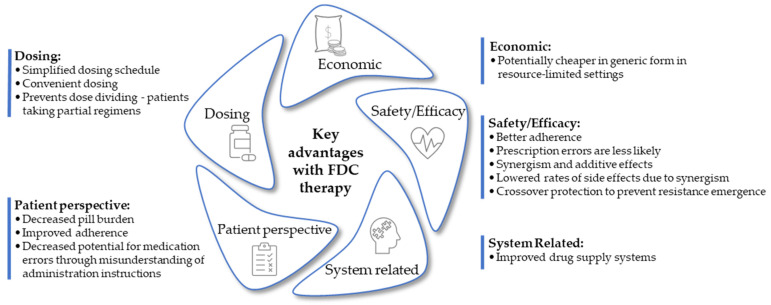
Pentagonal framework of identified advantages of fixed-dose combination therapy [34,37,38].

**Figure 2 pharmaceutics-16-00178-f002:**
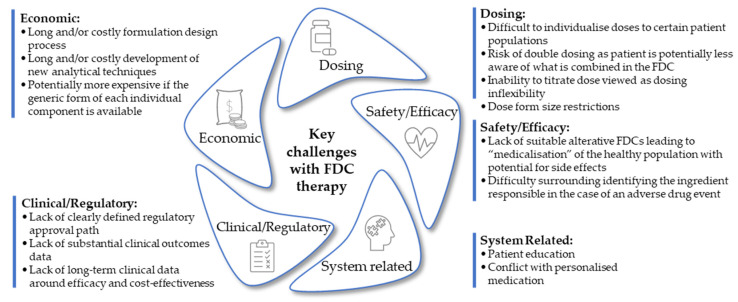
Pentagonal framework of identified barriers opposing fixed-dose combination therapy [34,37,38].

**Figure 3 pharmaceutics-16-00178-f003:**
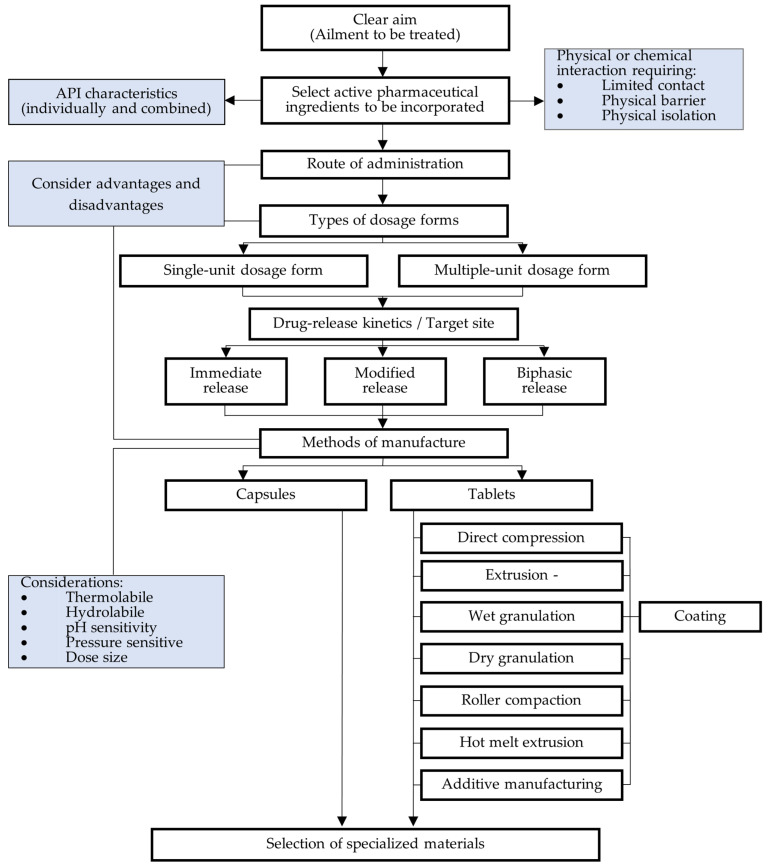
Schematic representation of the considerations taken during the dosage form design process of a fixed-dose combination product [111,115].

**Table 1 pharmaceutics-16-00178-t001:** Number of tablets administered using individual components versus fixed-dose combination products for first-line treatment of drug-susceptible TB in over 8 years of age with a pre-treatment weight of 50–70 kg [5,28].

Individual Therapeutic Components	Number of Tablets Daily	Fixed-Dose Combination (FDC)	Number of Tablets Daily
**Intensive phase: scenario one**			
Rifampicin (600 mg)	1	Rifampicin + Isoniazid + Pyrazinamide + Ethambutol (150/75/400/275 mg)	4
Isoniazid (300 mg)	1		
Pyrazinamide (500 mg)	3		
Ethambutol (400 mg)	2		
**Total:**	**7**	**Total:**	**4**
**Intensive phase: scenario two**			
Rifampicin (150 mg)	4	Rifampicin + Isoniazid + Pyrazinamide + Ethambutol (150/75/400/275 mg)	4
Isoniazid (100 mg)	3		
Pyrazinamide (500 mg)	3		
Ethambutol (400 mg)	2		
**Total:**	**12**	**Total:**	**4**
**Maintenance phase: scenario one**			
Rifampicin (600 mg)	1	Rifampicin + Isoniazid (300/150 mg)	2
Isoniazid (300 mg)	1		
**Total:**	**2**	**Total:**	**2**
**Maintenance phase: scenario two**			
Rifampicin (150 mg)	4	Rifampicin + Isoniazid (300/150 mg)	2
Isoniazid (100 mg)	3		
**Total:**	**7**	**Total:**	**2**

**Table 2 pharmaceutics-16-00178-t002:** Fixed-dose combination products for the prevention of cardiovascular disease that adhere strictly to the original polypill definition (adapted from [14,25,37,64]).

Brand Name(s)	Active Pharmaceutical Ingredients	Indication
CV-Pill kit (Torrent Pharmaceuticals)	Metoprolol, ramipril, aspirin, atorvastatin 50/5/75/10 mg	Primary prevention
Heart Pill (Excella Pharma)	Ramipril, aspirin, atorvastatin 2.5/100/4; 5/100/40; 10/100/40 mg	Primary prevention
Polycap™ (Cadila Pharmaceuticals)	Atenolol, hydrochlorothiazide, ramipril, aspirin, simvastatin 50/12.5/5/100/20 mg	Primary prevention
Polypill-E (Alborz Darou Pharmaceuticals)	Enalapril, hydrochlorothiazide, aspirin, atorvastatin 2.5/12.5/81/20 mg	Primary prevention
Polypill-V (Alborz Darou Pharmaceuticals)	Hydrochlorothiazide, valsartan, aspirin, atorvastatin 12.5/40/81/20 mg	Primary prevention
Polytorva^®^ (USV)	Ramipril, aspirin, atorvastatin 10/75/5 mg	Secondary prevention
Ramitorva™ (Zydus Cadila Healthcare)	Ramipril, aspirin, atorvastatin 5/75/10 mg	Secondary prevention
Red Heart Pill 1 (Dr Reddy’s Laboratories)	Atenolol, lisinopril, aspirin, simvastatin 50/10/75/40 mg	Secondary prevention
Red Heart Pill 2 (Dr Reddy’s Laboratories)	Lisinopril, hydrochlorothiazide, aspirin, simvastatin 10/12.5/75/40 mg	Secondary prevention
Starpill (Cipla)	Atenolol, losartan, aspirin, atorvastatin 50/50/75/10 mg	Secondary prevention
Trinomia^®^ (Ferrer)	Ramipril, aspirin, atorvastatin 2.5/100/40; 5/100/40; 10/100/40 mg	Secondary prevention
ZYCAD-4 kit (Zydus Cadila Healthcare)	Ramipril, aspirin, atorvastatin, metoprolol 5/75/100/50 mg	Secondary prevention

**Table 3 pharmaceutics-16-00178-t003:** Fixed-dose combination products approved for use in cardiovascular disease prevention that deviate from the traditional cardiovascular disease polypill components.

Brand Name(s)	Active Pharmaceutical Ingredients	Classification
Acesyl Co^®^ (Akacia); Ariprel Plus^®^ (Watson Pharma); Coversyl^®^ Plus (Servier); Pearinda Plus^®^ (Pharma Dynamics); Perindopril Co Unicorn^®^; Prexum Plus^®^ (Biogaran); Vectoryl Plus^®^ (Aspen)	Perindopril, indapamide 4/1.25 mg	ACE inhibitor + diuretic [28]
Accumax Co^®^ (Pfizer); Accuretic^®^ (Pfizer); Adco-Quinaretic^®^	Quinapril, hydrochlorothiazide 10/12.5; 20/12.5; 20/25 mg	ACE inhibitor + diuretic [28]
Aldazide (Pfizer)	Spironolactone, isobutyl hydrochlorothiazide 25/2.5 mg	Aldosterone antagonist + diuretic [28]
Altoran^®^-CH (Alembic); Arbozil™-CT (Zuventus); Asar^®^-CT (Glenmark); Edarbyclor^®^ (Takeda Pharmaceuticals; Valeant); Myotan^®^-CT (Synokem); Tezihart-CH (Leeford)	Azilsartan, chlorthalidone 40/12.5; 40/25 mg	Angiotensin-receptor blocking + thiazide-like diuretic [28]
Amilorectic^®^ (aspen); Moduretic^®^ (MSD); Adco-retic^®^, Betaretic^®^ (Ranbaxy)	Amiloride, hydrochlorothiazide 5/50; 2.5/25 mg	Potassium sparing agents + diuretic [28]
Amlodipine and benazepril (Dr Reddy’s Laboratories); amlodipine and benazepril (Watson); Benidep (Johnlee); benazepril and amlodipine (Systopic Laboratories)	Benazepril, amlodipine 10/2.5; 10/5; 20/5; 20/10; 40/5; 40/10 mg	ACE inhibitor + calcium-channel blockers [28]
Atacand Plus^®^ (AstraZeneca)	Candesartan, hydrochlorothiazide 16/12.5; 32/12.5; 32/25 mg	Angiotensin II antagonist + diuretic [28,62]
Atamra CV kit (Amra Remedies)	Atorvastatin, ramipril, clopidogrel 10/5/75 mg	HMG CoA reductase inhibitor, ACE inhibitor, platelet aggregation inhibitor [14]
Caduet (Pfizer)	Amlodipine, atorvastatin 5/10; 5/20; 5/40; 5/80; 10/10; 10/20; 10/40; 10/80 mg	Calcium channel blocker + HMG CoA reductase inhibitor [28]
Cibadrex^®^ (Novartis)	Benazepril, hydrochlorothiazide 10/12.5 mg	ACE inhibitor + diuretic [28]
Coaprovel^®^ (Sanofi-Aventis); Co-Irbewin^®^ (Withrop); Isart Co^®^ (Zydus)	Irbesartan, hydrochlorothiazide 150/12.5; 300/12.5 mg	Angiotensin II antagonist + diuretic [28]
Co-Renitec^®^ (MSD), Enap-Co^®^ (Pharma Dynamics), Pharmapress Co^®^ (Aspen)	Enalapril, hydrochlorothiazide 20/12.5 mg	ACE inhibitor + diuretic [28]
Cozaar Comp^®^ (MSD); Lohype Plus^®^ (Ranbaxy Betabs); Losacar Co^®^ (Zydus); Ciplazar Co^®^ (Cipla)	Losartan, hydrochlorothiazide 50/12.5 mg	Angiotensin II antagonist + diuretic [28]
Co-Pritor^®^ (Ingelheim); Co-Micardis^®^ (Ingelheim)	Telmisartan/hydrochlorothiazide 40/12.5; 80/12.5 mg; 80/25 mg	Angiotensin II antagonist + diuretic [28]
Co-Diovan^®^ (Novartis); Co-Tareg^®^ (Novartis); Co-Zomevek^®^ (Novartis)	Valsartan/hydrochlorothiazide 80/12.5; 160/12.5; 160/25 mg; 320/12.5; 320/25 mg	Angiotensin II antagonist + diuretic [28]
Co Exforge^®^ (Novartis)	Losartan, amlodipine, hydrochlorothiazide 160/5/212.5/; 160/10/12.5; 160/5/25; 160/10/25 mg	Angiotensin II antagonist + calcium-channel blocker [28]
Dyazide^®^ (Litha Pharma); Renezide^®^ (Aspen)	Triamterene, hydrochlorothiazide 50/25 mg	Potassium-sparing agent + diuretic [28]
Exforge^®^ (Novartis)	Valsartan, amlodipine 160/5; 320/5; 160/10; 320/10 mg	Angiotensin II antagonist + calcium-channel blocker [28]
Entresto™ (Novartis); Vymada^®^ (Novartis); Valsa 50 (Natco); Azmarda^®^ (Cipla); Valcubit^®^ (Elder)	Sacubitril, valsartan 24/26; 49/51; 97/103 mg	Neprilysin inhibitor + angiotensin II antagonist [28]
Fosinopril and hydrochlorothiazide (Biogaran^®^; Cipla; Citron; Glenmark; Ranbaxy; Rising^®^)	Fosinopril, hydrochlorothiazide 10/12.5; 20/12.5 mg	ACE inhibitor + diuretic [28]
Fortzaar^®^ (MSD); Lohype Forte Plus^®^	Losartan, hydrochlorothiazide 100/25 mg	Angiotensin II antagonist + diuretic [28]
Imprida^®^ HCT (Novartis)	Amlodipine, valsartan, hydrochlorothiazide 5/160/12.5; 10/320/25 mg	Calcium channel blocker + angiotensin II antagonist + diuretic [14]
Inhibace^®^ Plus (Roche)	Cilazapril, hydrochlorothiazide 5/12.5 mg	ACE inhibitor + diuretic [28]
Livper-A (Livealth); Perindopril- amlodipine-Mepha^®^ (Mepha); Perindopril- amlodipine-STADA (Stada); Perindopril- amlodipine teva (Teva)	Perindopril, amlodipine 3.5/2.5; 7/5; 14/10 mg	ACE inhibitor + calcium-channel blockers [28]
Losaar Plus^®^ (Accord); Sartoc Co (Aspen); Zartan Co^®^ (Pharma Dynamics); Hytenza Co^®^ (Watson Pharma); Netrasol^®^ (Specpharm)	Losartan, hydrochlorothiazide 50/12.5; 100/25 mg	Angiotensin II antagonist + diuretic [28]
Preterax^®^ (Servier)	Perindopril, indapamide 2/0.625 mg	ACE inhibitor + diuretic [28]
Polypill (Cipla)	Amlodipine, losartan, hydrochlorothiazide, simvastatin 2.5/25/12.5/40 mg	Calcium-channel blocker, angiotensin II antagonist, diuretic, HMG CoA reductase inhibitor [25]
RIL–AA (East West Pharma)	Ramipril, atorvastatin, aspirin 5/10/75 mg	ACE inhibitor, HMG CoA reductase inhibitor, platelet aggregation inhibitor [64]
Servatrin^®^ (Aspen)	Timolol, amiloride, hydrochlorothiazide 10/2.5/25 mg	β-blocker, non-selective, potassium-sparing agents + diuretic [28]
Spec-Perindopril Plus^®^	Perindopril, indapamide 2/0.625; 4/1.25 mg	ACE inhibitor + diuretic [28]
Tarka^®^ (Abbott)	Trandolapril, verapamil 2/180; 4/240 mg	ACE inhibitor + calcium-channel blockers [28]
Tenoretic^®^ (AstraZeneca); Sandoz Co-tenidone^®^ 100/25, 50/12.5; Tenchlor^®^ (Aspen); Tenoret 50^®^ (AstraZeneca); Tenchlor HS^®^ (Aspen)	Atenolol, chlortalidone 100/25; 50/12.5 mg	β-blockers, selective + diuretic [28]
Tri-Plen^®^ (Sanofi-Aventis)	Ramipril, felodipine 2.5/2.5; 5/5 mg	ACE inhibitor + calcium-channel blockers [28]
Triplixan^®^ (Servier)	Amlodipine; perindopril, indapamide 5/5/1.25; 10/10/2.5 mg	Calcium channel blocker + ACE inhibitor + diuretic [14]
Tritace Plus^®^ (Sanofi-Aventis)	Ramipril, hydrochlorothiazide 2.5/12.5; 5/12.5; 10/25 mg	ACE inhibitor + diuretic [28]
Triveram^®^ (Servier)	Perindopril, amlodipine, atorvastatin 10/40/10 mg	ACE inhibitor, calcium-channel blocker, HMG CoA reductase inhibitor [14]
Twynsta^®^ (Ingelheim)	Telmisartan, amlodipine 40/5; 80/5; 40/10; 80/10 mg	Angiotensin II antagonist + calcium-channel blocker [28]
Uniretic^®^ (Schwarz Pharma’s)	Moexipril, hydrochlorothiazide 7.5/12.5; 15/12.5; 15/25 mg	ACE inhibitor + diuretic [28]
Zaneril^®^ (Litha Pharma)	Enalapril, lercanidipine 10/10; 20/10 mg	ACE inhibitor + calcium-channel blockers [28]
Zapto-Co^®^ (Aspen)	Captopril, hydrochlorothiazide 50/12.5 mg	ACE inhibitor + diuretic [28]
Zestoretic^®^ (AstraZeneca); Auro-Lisinopril Co^®^ (Actor Pharma); Hexal-Lisinopril Co^®^ (Sandoz); Lisinopril Co Unicorn^®^; Lisoretic^®^ (Pharma Dynamics); Lisinozide^®^ (Novagen); Lisozide (Austell); Diace Co^®^ (Simayla); Zestozide^®^ (Mylan)	Lisinopril, hydrochlorothiazide 10/12.5; 20/12.5; 20/25 mg	ACE inhibitor + diuretic [28]
Ziak^®^ (Merck); Bilocor Co^®^ (Pharma dynamic); Bisoprolol Hydrochlorothiazide Zydus^®^ (Zydus)	Bisoprolol, hydrochlorothiazide 2.5/6.25; 5/6.25; 10/6.25 mg	Beta-blockers (selective) + diuretic [28]

**Table 4 pharmaceutics-16-00178-t004:** Examples of antibacterial fixed-dose combination products [28].

Brand Name(s)	Active Pharmaceutical Ingredients	Classification
Macron^®^ (Mylan); Megapen^®^ (Aspen)	Amoxicillin, flucloxacillin 250/250 mg	Beta-lactam sensitive penicillins with extended spectrum
Apen^®^ (Mylan)	Ampicillin, cloxacillin 250/250 mg	
Augmentin^®^ (Aspen); Adco-amoclav^®^ (AI Pharm); Amoclan^®^ (Watson pharma); AugMaxil^®^ (Aspen); Ranclav^®^ (Ranbaxy); Clamentin^®^ (Mylan); Austell-Co-Amoxiclav^®^; Bindoclav^®^ (Actor Pharma)	Amoxicillin, clavulanic acid 250/125; 500/125; 875/125 mg	Beta-lactam inhibitors and penicillins
Bactrim^®^ (Roche); Adco-Co-Trimoxazole^®^; Lagatrim^®^ (Akacia); Nucotrim^®^ (GulfDrug); Cozole^®^ (Ranbaxy); Purbac^®^ (Aspen)	Trimethoprim, sulfamethoxazole 80/400 mg	Trimethoprim and sulphonamides

**Table 5 pharmaceutics-16-00178-t005:** Examples of fixed-dose combination products available for treatment of malaria.

Brand Name(s)	Active Pharmaceutical Ingredients	Classification
Coartem^®^ (Novartis), Artefan^®^ (Ajanta), Lumet (Cipla), Lumerax (IPCA Laboratories Ltd.), Combiart^®^ (Strides Arcolab Limited), Lumiter (Macleods), Komefan (Mylan), Riamet^®^ (Novartis)	Artemether, lumefantrine 20/120 mg	Artemisinin + fluorene [28,68]
Pyramax^®^ (co-developed by MMV and Shin Poong Pharmaceutical)	Artesunate, pyronaridine 60/180 mg	Artemisinin + benzonaphthyridine derivative [69]
MEFLIAM (Cipla); Wellcigo Plus (Wellona Pharma); Falcigo Plus (Zydus Cadila)	Artesunate, mefloquine 25/50; 100/200 mg	Artemisinin + analogue of quinine [68]
Coarsucam™ (Sanofi-Aventis); Artesunate/Amodiaquine Winthrop^®^ tablet	Artesunate, amodiaquine 25/67.5; 50/135; 100/270 mg	Artemisinin + quinoline [68]
Artecospe^®^ (Guilin Pharmaceutical)	Artesunate, sulfadoxine, pyrimethamine 50/500/25 mg	Artemisinin + sulfonamide + folic acid antagonist
Fansidar^®^ (Akacia; Roche; Ascendis Pharma)	Sulfadoxine, pyrimethamine 250/12.5; 500/25 mg	Sulfonamide + folic acid antagonist [28,68]
Eurartesim (Alfasigma), Artekin (Holleykin), Duocotexin (Holley Pharm)	Dihydroartemisinin, piperaquine 10/80 mg	Artemisinin + aminoquinoline [68]
Malarone^®^ and Malanil™ (GlaxoSmithKline)	Atovaquone, proguanil hydrochloride 250/100 mg	Naphthoquinones + biguanide derivative [28]
Synriam^TM^ (Ranbaxy)	Arterolane maleate, piperaquine phosphate 150/750 mg	Adamantanes + aminoquinoline [70]

**Table 6 pharmaceutics-16-00178-t006:** Examples of first-line drugs for tuberculosis treatment formulated as a fixed-dose combination, single-tablet regimen.

Brand Name(s)	Active Pharmaceutical Ingredients	Indication
Isonarif ^TM^ (Versa Pharma); Rifamate^®^ (Sanofi); Rifinah (Sanofi-Aventis); Riwell-IS (Wellona pharma)	Rifampicin, isoniazid 150/75; 300/150 mg	First-line drugs in combination for tuberculosis treatment [5,28]
Rimactazid (Sandoz)	Rifampicin, isoniazid 150/75; 300/150; 60/60 mg	
Combunex 800 (Lupin); Ethox-IN (Talent healthcare)	Ethambutol, isoniazid 800/300 mg	
Rifater (Sanofi-Aventis); RismidCare (AdvaCare)	Rifampin, isoniazid, pyrazinamide 120/50/300 mg	
Onecure (TGP); Rifafour e-275 (Sanofi-Aventis)	Rifampicin, isoniazid, pyrazinamide, ethambutol 150/75/400/275 mg	

**Table 8 pharmaceutics-16-00178-t008:** Examples of antiretroviral fixed-dose combination products formulated in a single-tablet regimen for pediatrics and adolescents [28,35,100].

Brand Name(s) and Companies	Minimum Body Weight, Weight range, or age	Active Pharmaceutical Ingredients Classification				
		NRTIs/NtRTIs	NNRTIs	INSTI	PI	PK Enhancer
Cimduo^®^ (Mylan); Temixys™ (Janssen)	≥35 kg	Lamivudine (300 mg) Tenofovir disoproxil (300 mg)				
Combivir^®^ (GlaxoSmithKline, ViiV); Combozil (HeteroDrugs SA); Duovir (Cipla); Adco- lamivudine and zidovudine; Lamzid (Aspen); Loziv (Novagen Pharma)	30 kg	Lamivudine (150 mg) Zidovudine (300 mg)				
Descovy^®^ (Gilead)	14 to < 25 kg	Emtricitabine (120 mg) Tenofovir alafenamide (15 mg)				
	25–35 kg	Emtricitabine (200 mg) Tenofovir alafenamide (25 mg)				
Epzicom^®^ (US, ViiV), Kivexa^®^ (GSK, ViiV Healthcare); Dumiva (Mylan)	25 kg	Abacavir (600 mg) Lamivudine (300 mg)				
Truvada^®^ (Gilead); Adco-Emtevir (Adcock Ingrams); Tencitab (Aspen); Didivir (Cipla); Tyricten (Aurobindo); Tenemine (Mylan)	17 to <22 kg	Emtricitabine (100 mg) Tenofovir disoproxil (150 mg)				
	22 to <28 kg	Emtricitabine (133 mg) Tenofovir disoproxil (200 mg)				
	28 to <35 kg	Emtricitabine (167 mg) Tenofovir disoproxil (250 mg)				
	35 kg	Emtricitabine (200 mg) Tenofovir disoproxil (300 mg)				
Atripla^®^ (MSD); Atroiza (Mylan); Citenvir (Novagen); Odimune (Cipla); Tribuss™ (Aspen)	40 kg	Emtricitabine (200 mg) Tenofovir disoproxil (300 mg)	Efavirenz (600 mg)			
Complera^®^ (Aspen), Eviplera^®^ (Gilead)	35 kg and aged ≥12 years	Emtricitabine (200 mg) Tenofovir disoproxil (300 mg)	Rilpivirine (25 mg)			
Delstrigo^®^ (Merck & Co.)	35 kg	Lamivudine (300 mg) Tenofovir disoproxil (300 mg)	Doravirine (100 mg)			
Odefsey^®^ (Aspen)	35 kg and aged ≥12 years	Lamivudine (300 mg) Tenofovir disoproxil (300 mg)	Rilpivirine (25 mg)			
Symfi^®^ (Mylan); Tenarenz (Aspen)	40 kg	Lamivudine (300 mg) Tenofovir disoproxil (300 mg)	Efavirenz (600 mg)			
Biktarvy^®^ (Aspen)	14 to < 25 kg	Emtricitabine (120 mg) Tenofovir alafenamide (15 mg)		Bictegravir (30 mg)		
	25 kg	Emtricitabine (200 mg) Tenofovir alafenamide (25 mg)		Bictegravir (50 mg)		
Dovato (GlaxoSmithKline; ViiV Healthcare)	Minimum weight for individual components	Lamivudine (300 mg)		Dolutegravir (50 mg)		
Triumeq^®^ (GlaxoSmithKline)	25 kg	Abacavir (600 mg) Lamivudine (300 mg)		Dolutegravir (50 mg)		
Triumeq^®^ PD (GlaxoSmithKline)	10 to <25 kg	Abacavir (60 mg) Lamivudine (300 mg)		Dolutegravir (50 mg)		
Genvoya^®^ (Aspen)	25 kg	Emtricitabine (200 mg) Tenofovir alafenamide (10 mg)		Elvitegravir (150 mg)		Cobicistat (150 mg)
Genvoya^®^ (Aspen)	35 kg and SMR 4 or 5	Emtricitabine (200 mg) Tenofovir disoproxil (300 mg)		Elvitegravir (150 mg)		Cobicistat (150 mg)
Evotaz^®^ (Bristol-Myers Squibb)	35 kg				Atazanavir (300 mg)	Cobicistat (150 mg)
Prezcobix^®^ (Janssen)	>40 kg				Darunavir (800 mg)	Cobicistat (150 mg)
Kaletra^®^ (AbbVie)	>40 kg				Lopinavir (100 mg, 200 mg tablets)	Ritonavir (25 mg, 50 mg tablets)

INSTI, Integrase strand transfer inhibitors; NRTIs/NtRTIs, nucleoside and nucleotide reverse transcriptase inhibitors; NNRTIs, non-nucleoside reverse transcriptase inhibitors; PI, protease inhibitor; PK, pharmacokinetic; SMR, sexual maturity ratings relating to tenofovir disoproxil-associated bone toxicity.

**Table 9 pharmaceutics-16-00178-t009:** Examples of available fixed-dose combination therapies for the control of glycemic levels in diabetes mellitus type 2.

Brand Name and Company	Combination Drugs	Classification
Acarjohn-M (Johnlee Pharmaceuticals)	Acarbose, metformin 25/500 mg	Alpha-glucosidase inhibitors + biguanide [102]
Avandamet^®^ (GSK)	Rosiglitazone, metformin 2/500; 4/500; 2/1000; 4/1000 mg	Thiazolidinediones + biguanide [101,102]
Janumet^®^ (MSD)	Sitagliptin, metformin 50/500; 50/850; 50/1000 mg	Dipeptidyl peptidase 4 inhibitor + biguanide [28,101,102]
Amaryl^®^ (Sanofi-Aventis)	Glimepiride, metformin 1/500; 2/500 mg	Sulfonylureas + biguanide [28,101,102]
Glucovance^®^ (Merck)	Glibenclamide, metformin 1.25/500; 2.5/500; 5/500 mg	
Metaglip^TM^ (Bristol-Myers Squibb)	Glipizide, metformin 2.5/250; 2.5/500 mg	
Galvus Met^®^ (Novartis)	Vildagliptin, metformin 50/1000; 50/850 mg	Dipeptidyl peptidase 4 inhibitor + biguanide [28,101]
PrandiMet^®^ (Sciele and Novo Nordisk)	Repaglinide, metformin 1/500; 2/500 mg	Meglitinides + biguanide
Avandaryl^®^ (GlaxoSmithKline)	Rosiglitazone, glimepiride 4/1; 4/2; 4/4; 8/2; 8/4 mg	Thiazolidinediones + sulfonylureas [102]
Duetact^®^ (JPI, Takeda Pharmaceuticals)	Pioglitazone, glimepiride 30/2; 30/4 mg	
Jentadueto^®^ (Boehringer Ingelheim)	Linagliptin, metformin 2.5/850; 2.5/1000 mg	Dipeptidyl peptidase 4 inhibitors + biguanide [101]
Kazano (Takeda Pharmaceuticals)	Aloglipine, metformin 12.5/500; 12.5/1000 mg	
Komboglyze^®^ (AstraZeneca & Bristol-Myers Squibb)	Saxagliptin, metformin 2.5/500; 2.5/850; 2.5/1000 mg	
Invokamet^TM^ (Janssen Pharmaceuticals)	Canagliflozin, metformin 50/500; 50/850; 50/1000; 150/500; 150/850; 150/1000 mg	Sodium-glucose co-transporter 2 inhibitors + biguanide [101,102]
Xigduo^®^ (AstraZeneca)	Dapagliflozin, metformin 5/850; 5/1000 mg	
Synjardy^®^ (Boehringer Ingelheim)	Empagliflozin, metformin 5/500; 5/850; 5/1000; 12.5/500; 12.5/850; 12.5/1000 mg	
Segluromet^®^ (Pfizer)	Ertugliflozin, metformin 2.5/850; 2.5/1000; 7.5/850; 7.5/1000 mg	
Glyxambi^®^ (Boehringer Ingelheim)	Empagliflozin, linagliptin 10/5; 25/5 mg	Dipeptidyl peptidase 4 inhibitors + sodium- glucose co-transporter 2 inhibitors [101,102]
Qtern^®^ (AstraZeneca)	Saxagliptin, dapagliflozin 5/10 mg	
Steglujan^®^ (Merck)	Sitagliptin, ertugliflozin 5/100; 15/100 mg	

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
