# Peer review of "Fixed-Dose Combination Formulations in Solid Oral Drug Therapy: Advantages, Limitations, and Design Features"

_pharmaceutics, 2024, doi:10.3390/pharmaceutics16020178_

Round 1

Reviewer 1 Report

Comments and Suggestions for Authors

Dear authors,         

The manuscript presents an interesting and extensive review regarding the advantages and limitations of FDC. Please find below a few comments for your consideration:

·       Nowadays, antibiotic resistance represents a major health crisis and should be discussed. The association of antibiotics leads to synergism of action, reduces the risk of antibiotic resistance, etc. Thus, in the last years, many antibiotics have been associated (e.g. beta-lactam/beta-lactamase inhibitor, ceftolozane/tazobactam, ceftazidime/avibactam, meropenem/vaborbactam, and cilastatin-imipenem/relebactam) (please see: https://pubmed.ncbi.nlm.nih.gov/29671219/; https://www.mdpi.com/2227-9059/10/5/1121)

·       PDEPT (polymer-directed enzyme prodrug therapy), PELT, ADEPT etc. that uses fixed combination are promising new strategies in different pathologies such as oncology (eg.: Irinotecan and 5-FU co-loaded nanoparticles, doxorubicin–sorafenib co-loaded nanoparticles etc.)

https://www.mdpi.com/1999-4923/14/9/1773, https://www.ncbi.nlm.nih.gov/pmc/articles/PMC2375098/

·       The heading 5.1. Hypertension should be changed with CVD because the FDC presented is used in different cardiovascular pathologies, not only hypertension

·       Authors should include sacubitril and valsartan – FDC used in heart failure with a reduced ejection fraction as 1st line of therapy. https://pubmed.ncbi.nlm.nih.gov/30565021/

·       Please revise the number of headings: e.g. 5.2. Tuberculosis must be 5.3.

·       Other very used FDC should be presented: premixed insulins, analgesic - antipyretic drugs, anti-inflammatory drugs and PPI, oral contraceptives, etc. For example, the authors mention strategies regarding the combination of ibuprofen, but ibuprofen is not present as an example of FDC. The same for amoxicillin combinations.

·       The limitations of the study should be presented.

Author Response

Please see attached document for response to reviewer comments for Reviewer 1.  

Reviewer 2 Report

Comments and Suggestions for Authors

Manuscript ID: pharmaceutics

Title: Fixed-dose combination formulations in drug therapy: advantages, limitations, and design features

Authors: Christi A. Wilkins, Hannlie Hamman, Josias H. Hamman, Jan H. Steenekamp

The review „Fixed-dose combination formulations in drug therapy: advantages, limitations, and design features” concerns fixed-dose combination drugs (FDC), which are one of the ways to simplify pharmacotherapy. A combination medicine uses two or more active substances in a dosage unit, such as a capsule or tablet. The main assumption of combined therapy is to achieve a synergistic effect between the active substances. This synergism may be additive, when the ingredients used have the same attachment point and mechanism of action, or hyperadditive, when individual substances have different attachment points and mechanisms of action.

The benefits of FDC therapy are obvious, but the title covers both the advantages, limitations, and features of FDC, but omits future prospects. This is a key element of all review works, which, apart from the historical outline and the current situation, should indicate the direction of development, in this case of FDC therapy. Lipid-based formulations and three-dimensional printing, which are only mentioned in a few sentences, are not enough. Certenly, the work should be extended to include a future perspective.

The authors divided the article into chapters: (1) introduction, (2) FDC combinations in drug therapy, (3) advantages of FDC formulations in drug therapy, (4) challenges of FDC formulation in drug therapy, (5) conditions commonly treated with FDC formulations, and in it (5.1) hypertension, (5.2) malaria, (5.3 not 5.2) tuberculosis, (5.4) Human Immunodeficiency Virus, (5.5) diabetes mellitus [why did the authors refer only to these disease entities? while the pharmaceutical market is dominated by FDC drugs in the treatment of not only cardiovascular diseases (hypertension, hypercholesterolemia) and infectious diseases (Helicobacter pylori infection, HIV or tuberculosis), but also the treatment of endocrine and nervous system diseases (depression, Alzheimer's disease), respiratory diseases (asthma, chronic obstructive pulmonary disease) and allergies. Additionally, the Authors indicated that the purpose of the article is  „This article intends to critically review the use of FDC therapy and provide an insight into FDC products..” (lines 39-40) all FDC therapy or selected therapies?], (6, the font is bold, not italic as in previous sections titles) FDC formulation factors for consideration, (7) formulation approaches and (8) conclusion.

Subsection 4.4, development of analytical methods, does not contribute anything except that the development of selective analytical methods in the case of FDC is a complicated and time-consuming process. This is obvious, as is the fact that each registered medicinal product is obligatorily subject to evaluation of chemical-pharmaceutical-biological, toxicological and clinical documentation, and the methods must allow for the assessment of the quality of the medicinal product.

The 'Classification' column in Table 2 is not consistent with the 'Classification' column in Tables 3, 4 and 8, where the classification means therapeutic groups.

I have a few comments on the text. The first concerns the language in which the article was written. The text is written in quite a "heavy" language, although in its current form it is rather of a popular-scientific nature. The second concerns the insufficient emphasis on the novelty of the work. Most likely due to the lack of discussion of new, future solutions and challenges of FDC therapy.

Tables 6 and 7 refer to FDC antiretroviral products, while the text in lines 387–390 refers to cotrimoxazole, an antibacterial product that is not included in Tables 6 and 7. Similar lines 412-421 concern anti-tuberculosis drugs: rifampicin, isoniazid, pyrazinamide, in anti-HIV part products. Moreover, sample Table 4 contains nine FDCs, and below the table the description only concerns the pharmacodynamics of lumefantrine in the context of protection against Plasmodium species. It is unknown why only this element is described in the text?

Some issues were discussed briefly and quite generally. The sections are short, which is an advantage on the one hand, but on the other hand they do not allow for a deeper insight into the essence of the problem, which perpetuates the impression of a laconic treatment of the problem.

The conclusions should also change.

Author Response

Please see attached document for response to reviewer comments for Reviewer 2.  

Reviewer 3 Report

Comments and Suggestions for Authors

Readers of the Pharmaceutics will undoubtedly find the authors' review, which is focused on fixed-dose combination formulations in drug therapy, to be rather interesting. The review's advantages include the material's comprehensive presentation, a solid factual foundation, insightful graphics, and clear English.

I find the review to be very interesting and I would like to suggest that it be published after minor changes. What I would like to do is add an additional chapter, at least, on drugs delivery techniques that are now being developed. Talking about the potential for mixed dosage forms development would be highly suitable.

Moreover the manuscript needs to be accompanied with graphical abstract. 

Author Response

Please see attached document for response to reviewer comments for Reviewer 3.  

Round 2

Reviewer 2 Report

Comments and Suggestions for Authors

I accept the authors' responses. I recommend publishing the article after the changes introduced by the authors.